# A new airborne system for simultaneous high-resolution ocean vector current and wind mapping: first demonstration of the SeaSTAR mission concept in the macrotidal Iroise Sea

5 David. L. McCann[1], Adrien C. H. Martin[1,2], Karlus Macedo[3], Ruben Carrasco[4], Jochen Horstmann[4], Louis Marié[5], José Márquez-Martínez[6], Marcos Portabella[7], Adriano Meta[3], Christine Gommenginger[1], Petronilo Martin-Iglesias[8], Tania Casal[8]

[1]National Oceanography Centre, Liverpool, L55DA, United Kingdom
10 [2]NOVELTIS, Labège, 31670, France
[3]Metasensing BV, Noordwijk, 2201 DK, Netherlands
[4]Helmholtz-Zentrum Hereon, Geestacht, 21502, Germany
[5]Ifremer, Plouzané, 29280, France
[6]Radarmetrics SL, Santander, 39005, Spain
15 [7]Institut de Ciències del Mar, Barcelona, 08003, Spain,
[8]European Space Agency, European Space Research and Technology Centre, Noordwijk, 2201 AZ, Netherlands

*Correspondence to*: D. L. McCann (david.mccann@noc.ac.uk)

**Abstract.** Coastal seas, shelf seas and marginal ice zones are dominated by small-scale ocean surface dynamic processes that play a vital role in the transport and exchange of climate-relevant properties like carbon, heat, water and nutrients between land, ocean, ice and atmosphere. Mounting evidence indicates that ocean scales below 10 km have far-ranging impacts on air-sea interactions, lateral ocean dispersion, vertical stratification, ocean carbon cycling, and marine productivity – governing exchanges across key interfaces of the Earth System, the global ocean and atmosphere circulation and climate. Yet, these processes remain poorly observed at the fine spatial and temporal scales necessary to resolve them. The Ocean Surface Current Airborne Radar (OSCAR) is a new airborne instrument with the capacity to inform these questions by mapping vectorial fields of total ocean surface currents and winds at high resolution over a wide swath. Developed for the European Space Agency (ESA), OSCAR is the airborne demonstrator of the satellite mission concept 'SeaSTAR', which aims to map total surface current and ocean wind vectors with unprecedented accuracy, spatial resolution and temporal revisit across all coastal seas, shelf seas and marginal ice zones. Like SeaSTAR, OSCAR is an active microwave Synthetic Aperture Radar Along-Track Interferometer (SAR-ATI) with optimal three-azimuth sensing enabled by unique highly-squinted beams. In May 2022, OSCAR was flown over the Iroise Sea, France, in its first scientific campaign as part of the ESA-funded SEASTARex project. The campaign successfully demonstrated the capabilities of OSCAR to produce high-resolution 2D images of total surface current vectors and near-surface ocean vector winds, simultaneously, in a highly dynamic, macrotidal coastal environment. OSCAR current and wind vectors show excellent agreement against ground-based X-band radar derived surface currents, numerical model outputs and NovaSAR-1 satellite SAR imagery, with Root-Mean-Square differences against X-band radar better than 0.2 m s$^{-1}$ for currents at 200 m resolution. These results are the first demonstration of simultaneous retrieval of total current and wind vectors from a high-squint three-look SAR-ATI instrument, and the first geophysical validation of the OSCAR and SeaSTAR observing principle. OSCAR presents a remarkable new ocean observing capability to support the study of small-scale ocean dynamics and air-sea interactions across the Earth's coastal, shelf and polar seas.

# 1 Introduction

The Earth's coastal and shelf seas and marginal ice zones are dominated by dynamic surface processes that exist on much shorter spatial and temporal scales than in the open ocean (Schulz-Stellenfleth and Stanev, 2016; Kozlov et al., 2019). These small-scale dynamics (order of 1–10 km) play a key role in the exchange and transport of essential ocean and climate variables, biogeochemical processes, vertical mixing and air-sea interactions (Martin and Richards, 2001; Levy et al., 2012; Sasaki et al., 2014), however global observations of dynamics at these scales are rare (McWilliams, 2016). Ocean surface currents and winds and their resultant air-sea interactions show increased complexity in coastal seas (Bricheno et al., 2018), shelf seas and in the vicinity of sea ice. The limited spatial resolution of wind predictions, model forcing and validation data have direct impacts on the ability to predict storms (Mass et al., 2002; Maskell, 2012) and assess wind power resources in developing areas (Ruiz et al., 2022). Wind observations are particularly lacking in coastal seas where satellite scatterometry is unable to make useful measurements within ~10 km of the coast (Martin et al., 2018). Simultaneous measurements of total current and

wind vectors at fine scale are needed to improve understanding and predictive capability of coastal and shelf sea processes (Samleson, 2019; Villas Bôas et al., 2019; Hauser et al., 2023). With ocean prediction systems aspiring to resolve small scale processes operationally across the Earth's coastal, shelf and polar seas by the end of the decade (Holt et al., 2017), high-resolution mapping of these key physical properties will become increasingly important to validate and improve the numerical

models used for short-term forecasting and climate change projections. There is therefore a clear and present need for synoptic measurements of currents and winds to meet the scientific, economic and societal challenges of the future.

SeaSTAR is a satellite mission concept submitted to the European Space Agency (ESA) Earth Explorer program to study these fast-changing, small scale ocean surface dynamics across the world's coastal oceans, shelf seas and marginal ice zones (Gommenginger et al., 2019). SeaSTAR is based on the principle of Synthetic Aperture Radar Along-Track Interferometry

(SAR-ATI) applied in three highly squinted directions in azimuth. SAR-ATI measures the Doppler shift of ocean microwave backscatter signals from pairs of complex SAR images separated by a short time lag. The measured Doppler shift (interferogram) relates to the motion of the ocean surface in the radar line-of-sight, which contains the total ocean surface current and the motion of microwave scatterers on the ocean surface (known as the wave Doppler or Wind-wave induced Artefact Surface Velocity 'WASV'; Martin et al., 2016). The total surface current vector (TSCV), corresponding to the

effective mass transport, contains contributions from surface wind drift, Ekman currents, geostrophic currents and Stokes drift but is dominated by tidal currents in many coastal environments by an order of magnitude. TSCV is derived from the surface motion directly measured by the Doppler shift minus the WASV.

SAR-ATI has a long heritage, from the first broadside-only ATI measurements of ocean surface velocity by Goldstein and

Zebker (1987), through to the two-look 'Wavemill' concept (Buck, 2005) and the SeaSTAR three-look concept (Martin et al., 2018; Gommenginger et al., 2019). To date, no three-look SAR-ATI system has ever been flown in space. SeaSTAR represents a ground-breaking ocean observing system that would apply the cutting-edge performance and spatial resolution of SAR-ATI to deliver simultaneous fine-scale images of total surface current and wind vectors to inform these vital but poorly observed ocean processes.

The Ocean Surface Current Airborne Radar (OSCAR) instrument is a unique, 3-look SAR-ATI, airborne system operating at Ku-band (13.5 GHz). Built and operated by MetaSensing BV in Italy and the Netherlands, OSCAR was developed for ESA to demonstrate the three-look SAR-ATI principle and the ability to simultaneously observe total surface current vectors (TSCV) and ocean surface vector winds (OSVW) in two dimensions, at high accuracy and fine resolution, in a single pass, from an airborne platform (Trampuz et al., 2018). Within ESA Earth Explorer 11 Phase 0 activities, the SEASTARex project performed

OSCAR airborne campaigns to increase the scientific readiness level of the SeaSTAR mission concept and act as the first scientific demonstration and validation of the OSCAR instrument and the high-squint 3-look SAR-ATI capability. Involving airborne OSCAR acquisitions, in situ observations, ground-based remote sensing, numerical modelling and earth observation, SEASTARex drew together expertise from nine international scientific institutions and specialists in the field of oceanography, airborne and ground-based remote sensing, engineering and earth observation sciences. This paper presents the activities and

results from the scientific airborne campaign of the OSCAR instrument over the Iroise Sea in May 2022, representing the first instance of simultaneous retrieval of total surface current vectors and ocean surface vector winds based on experimental measurements from a high-squint three-look airborne SAR-ATI instrument.

## 2 OSCAR airborne system and data

### 2.1 OSCAR instrument

OSCAR is a unique, high-squint, 3-look SAR-ATI instrument operating at Ku-band (13.5 GHz), featuring two pairs of interferometric antenna squinted at ±45° ('fore' and 'aft') from the aircraft broadside, and a conventional 'zero-Doppler' antenna pointing broadside ('mid'). All antennas pointed to the port (left) side of the aircraft and transmitted and received in vertical polarisation ('VV'). The instrument is mounted on a 3-axis gimbal (Fig. 1a) with a mounted Inertial Measurement Unit (IMU) paired with high-precision Global Navigation Satellite System (GNSS) receiver to stabilize the instrument pointing relative to the aircraft and resolve the pointing of the beams to a high precision. The antennas and gimbal are mounted in a purpose-built radome specifically designed for interferometry at Ku-band and Ka-band frequencies. OSCAR was installed on a PA-31 Piper Navajo airframe owned and flown by Metasensing BV (Figure 1b).

### 2.2 Data processing: simultaneous current and wind vector retrieval

The simultaneous retrieval of total surface current vector (TSCV) and ocean surface vector wind (OSVW) was performed using the SeaSTAR project in Python (Martin et al., 2023), which is based on the simultaneous wind-current retrieval method of Martin et al. (2018), adapted for the OSCAR 3-look configuration. At the centre of the simultaneous wind-current retrieval method is the minimization of the cost function $J$, defined here as:

$$J(u_{10}, c) = \frac{1}{N_S + N_D} \sum_{i=1}^{N_S} \left( \frac{KuMod(u_{10}, \chi_i, \theta_i, p_i) - \sigma^0_{obs,i}}{\Delta \sigma^0_i} \right)^2 + \frac{1}{N_S + N_D} \sum_{i=1}^{N_D} \left( \frac{KuDop(u_{10}, \chi_i, \theta_i, p_i) + c_{||_i} - RSV_{obs,i}}{\Delta RSV_i} \right)^2 \quad (1)$$

where $i$ is the beam index in a given azimuth direction (fore, mid, aft), $\sigma^0_{obs,i}$ is the observed Normalised Radar Cross Section (NRCS) in VV polarization for the beam index i, $RSV_{obs,i}$ is the measured Radial Surface Velocity for the beam index $i$ (in this case only fore and aft), $u_{10}$ is the stress-equivalent OSVW at 10 m height (de Kloe et al., 2017), $c$ is TSCV, $N_S$ is the total number of observations for NRCS, $N_D$ is the total number of observations for RSV, $KuMOD$ is the predicted NRCS using the chosen Geophysical Model Function (GMF) for Ku-band, $KuDOP$ is the predicted RSV using the chosen GMF for Ku-band

WASV, $\chi$ is the azimuth look direction, $\theta$ is the incidence angle from nadir, $p$ is the radiometric polarisation (in this case VV) and $c_{\|_i}$ is the component of TSCV along azimuth look direction for beam index $i$.

It should be noted that aside from the adaption for three simultaneous looks, Equation 1 differs from Martin et al (2018), specifically in the definition of the wind vector. In Martin et al (2018), $u_{10} = u_{ERW}$ is defined as the Earth relative wind (same as for the Numerical Weather Prediction NWP product), in this paper $u_{10} = u_{OSVW}$ is the stress equivalent Ocean Surface Wind at 10 m height which is equivalent to the difference between the Earth relative wind and the surface current, i.e. $u_{OSVW} = u_{ERW} - c$. Despite $u_{ERW}$ being the wind used in numerical weather prediction, the GMFs for $\sigma^0$ and Doppler shift used are at first order functions of $u_{OSVW}$ and not $u_{ERW}$, hence this slight modification.

For the results presented in this paper the $KuDOP$ GMF used for Equation 1 is an adapted version of the Envisat-ASAR C-band (5.4 GHz) empirically derived 'C-DOP' GMF (Mouche et al. 2012). The RSV derived from this model has proven to vary little with radar frequency and to be applicable over a wide range with examples in 9.8 GHz X-band (Martin et al., 2016; Elyouncha et al., 2024) up to 35.7 GHz Ka-band (Yurovsky et al., 2019; Polverari et al., 2022). The range of incidence angle for C-DOP is here extended up to 62° (empirically derived only up to 44°). These assumptions will be discussed in Section 5. Here C-DOP has been adapted in two ways: firstly, for Ku-Band use via a frequency shift of the Doppler calculation and secondly, extending the model's applicable upper range of incidence angle from 44° to 62°. For the $KuMOD$ GMF the NSCAT-4DS model (Wang et al., 2017) is used.

The parameters $\Delta\sigma^0 = k_p\sigma^0$ (where $k_p$ is the radiometric resolution) and $\Delta RSV$ represent the uncertainties in the measurements. Here $k_p$ was estimated as a combination of contributions from instrument noise and geophysical noise (Mejia et al., 1999; Portabella and Stoffelen, 2006; Anderson et al, 2012). Instrument noise was estimated via a robust estimator of distribution (normalized interquartile range) of measured $\sigma^0$ for each look direction. Geophysical noise was estimated via comparison of measured $\sigma^0$ with the predicted NRCS using NSCAT-4DS. $\Delta RSV$ was estimated using open-ocean OSCAR data from the SEASTARex campaign and ground truth data from an Acoustic Doppler Current Profiler (ADCP) mooring at 48.256°N, 5.249°W to estimate the observed WASV and comparing this to the predicted WASV from Mouche et al. (2012). The estimated noise parameters used in this study were $k_p$ = 20% and $\Delta RSV$ = 0.2 m s⁻¹.

The cost function $J$ is a unit-less function of 4 unknown variables ($\overrightarrow{u_{10}}, \vec{c}$). Minimising the cost function finds the values of TSCV (c) and OSVW ($u_{10}$) that best reduce the quadratic differences between the measured observables ($\sigma^{0,}_{obs,i}$, $RSV_{obs,i}$) and the predicted quantities ($KuMod$, $KuDop$). As in scatterometry, the minimisation returns up to four solutions (Portabella et al., 2002; Martin et al., 2018), leading to a well-known ambiguity problem and the requirement for an ambiguity removal procedure. A usual procedure to remove these ambiguities is to use additional geophysical information, e.g., from wind

forecasts (Portabella and Stoffelen, 2004). In the case of the work presented within this paper a simple ambiguity removal method was implemented, selecting the solution for $J$ closest in $\overrightarrow{u_{10}}$ space to wind vectors taken from the MeteoFrance

operational Application of Research to Operations at Mesoscale (AROME) atmospheric wind model (Seity et al., 2011).

## 3. Iroise Sea airborne campaign

The SEASTARex airborne campaign was conducted between the 17[th]–26[th] May, 2022 over the Iroise Sea, west of Brest, France, from the home airport of Morlaix, Brittany. Macrotidal and relatively shallow, the Iroise Sea is dominated by strong tidal currents and prevailing Atlantic swell interacting with complicated coastline morphology and bathymetry of the area

(Muller et al., 2009). The island of Ushant (Ouessant in French) and its coastal waters, at the westerly end of the Molène archipelago in the Iroise sea, experience some of the fastest tidal flows on the North European shelf, often exceeding 3 m.s$^{-1}$ (Sentchev et al., 2013) and frequent westerly and south-westerly Atlantic storms. Figure 1c shows the location of the study area and the bathymetry (in metres below mean sea level) from the European Marine Observation and Data Network (EMODNet) harmonized digital terrain model.

The airborne scientific campaign consisted of four flight days of repeat acquisitions of OSCAR data over three main areas of interest: a site around the Island of Ushant with highly dynamic, macrotidal and hydrodynamically heterogenous currents; a site south of La Jument lighthouse (Fig. 1d), with temporally and spatially homogenous conditions and deployed in situ buoy measurements (48.256°N, 5.249°W); and a long, open-ocean flight further south to coincide with measurements from the ASCAT satellite scatterometer. This paper focuses on the results from the OSCAR flights over the dynamic tidal race around

Ushant (flight tracks from Figure 1d) on the 17[th] and 22[nd] of May 2022 .

### 3.1 OSCAR data acquisitions

The OSCAR flights were scheduled to occur at an altitude above sea level of 3,000 m for all days, however a low cloud ceiling on the 17[th] May forced the aircraft to acquire data at an altitude of 1,950 m. Acquisitions on the 22[nd] may were obtained at the scheduled altitude of 3,050 m. All flights were performed at a mean ground speed of 80 m s$^{-1}$. The flights over Ushant were

scheduled to capture peak tidal flows, with the aircraft passing over the island at 09:32 UTC on the 17[th] and 05:48 UTC on the 22[nd] May. OSCAR was configured to generate SAR imagery at 8 m pixel resolution in a 5 km swath, with incidence angles varying between 22–69° for the fore and aft squinted channels and between 16–61° for the mid channel. Level 0 processing (e.g. SAR focusing) and radiometric and interferometric calibration of the OSCAR acquisitions were performed by Metasensing BV and Radarmetrics SL. Radiometric calibration was performed via targeted flights over corner reflectors of a

known radar cross section, with these data being recorded on each flight day before the scientific acquisitions took place. Additional flights over the corner reflectors were performed at the end of each flight to check that calibration parameters had not changed. Additional residual calibration of NRCS was performed using OSCAR data from the open ocean flights and computing the incidence-angle dependent bias with respect to NSCAT-4DS. Interferometric calibration was performed using over-land OSCAR data to assess the recorded Doppler velocities of static land reflections (which should theoretically be zero).

Level-1 single-look complex Doppler images were spatially smoothed using a 7-pixel (56 m) windowed mean and down-sampled to 200 m ground resolution (using the mean value in each 200 m cell) for input to the simultaneous retrieval. This resolution was chosen as a trade-off between capturing sub-mesoscale hydrodynamic features and computation time for the cost function. All campaign data is classified according to its stage in the processing chain, with Level-1 (L1) data corresponding to SAR-processed data in instrument (i.e., local) coordinates, Level-1c (L1c) data corresponding to calibrated

data on a shared grid and Level-2 (L2) data corresponding to retrieved geophysical parameters in a global coordinate system.

## 3.2 X-band marine radar

    As part of the SEASTARex project, a marine radar was installed on La Jument lighthouse overlooking the tidal race to the south-west of Ushant, which is an ideal location for observing the extreme tidal dynamics of the area (Filipot et al., 2019). The

coherent-on-receive X-band (9.3 GHz) radar (Horstmann et al., 2021) was mounted on the lighthouse at a height of 48 m above mean sea level and covered a radius between 52 m and 3150 m corresponding to grazing angles between 40° and 0.78° (equal to incidence angles of 50° and 89.22° respectively). During the campaign the radar was operated in its rotational mode, acquiring radar backscatter intensity and radial Doppler velocity maps with a repetition rate of approximately 0.5 Hz. All radar data were collected at a pulse length of 0.5 ns and with a pulse repetition frequency of 2 kHz at VV polarization with a 2.3 m

slotted waveguide antenna, resulting in an azimuthal resolution of 1.2°. Radar data where acquired at 20 MHz, resulting in a range resolution of 7.5 m. The radar was operated on 17[th] May 2022 between 6:41–12:00 UTC collecting six 10-minute video sequences every hour, which were used to compute surface current fields. One of these sequences was coincident with an OSCAR flight overpass of the radar at 09:38 UTC.

The surface current maps are derived from the wave signal within a 10-minute marine radar image sequence which is then transformed from the spatial-temporal domain to the wave number-frequency domain by a 3D Fast Fourier Transform (FFT). Within this 3D radar image spectrum, the wave signal is located on an inverted cone; the so-called dispersion shell defined by the linear dispersion relation of surface waves. In the case of deep water with respect to the wavelength this dispersion shell is solely dependent on the surface current (Senet at al., 2001; Huang et al., 2016). The 2D fields of current vectors are determined

by an algorithm that searches for the current that maximizes the energy on the dispersion shell using a brute-force optimization algorithm (Streßer et al., 2017) and considering wave lengths between 15–125 m and wave periods of between 4–20 s. The spatial window used for one individual current measurement spans over an area of 500 x 500 m, and the individual windows overlap by 200 m, which corresponds to the pixel ground resolution (posting) in the resulting 2D current map.

## 3.3 Numerical models

Data from two numerical models were used: the MARS2D depth-averaged hydrodynamic model (Lazure and Dumas, 2008) run operationally by the Laboratory for Ocean Physics and Satellite remote sensing (LOPS), and the MeteoFrance operational

AROME wind model (Seity et al., 2011). MARS2D depth-average currents were provided at 15-minute intervals at a ground resolution of 250 m. Hourly AROME wind forecasts were provided for an altitude of 10 m ($u_{10}$) at a ground resolution of 0.025° (approx. 2 km).

### 3.4 Spaceborne SAR imagery

NovaSAR-1 is a UK-funded technology demonstration satellite owned and operated by Surrey Satellite Technology Ltd. (SSTL) that delivers high-resolution S-band SAR imagery . NovaSAR-1 was commissioned to acquire images over the study site, resulting in an overpass at 10:30 UTC on 17th May 2022, coinciding with the OSCAR flight over Ushant with only an hour difference. The NovaSAR-1 data were commissioned and accessed thanks to the support of Martin Cohen at Airbus Defence and Space Ltd.

## 4 Results

### 4.1 OSCAR results for ebb tide on 17 May 2022

Figure 2 shows the retrieved TSCV (Fig. 2a) and OSVW (Fig. 2b) at 200 m ground resolution from a single OSCAR acquisition flying North (looking left) over the Island of Ushant during ebb tide at 09:38 UTC on the 17th May 2022. Sea state conditions at the time of acquisition consisted of a moderate swell from the south-west with a significant wave height of 2.5 m and a peak period of 11 s. Winds were approximately 7 m s$^{-1}$ from the south-east. The vectors on Figures 2a and 2b are plotted with a sub-sampled posting of 400 m for clarity. The trapezoidal shape of the OSCAR swath is due to the squinted look direction (±45° in azimuth) of the fore and aft ATI channels and their orthogonal combination to compute the L2 current and wind vectors. For comparison, Figure 2c shows depth-average current vectors at 250 m ground resolution (vector posting at 500 m) from the MARS2D ocean circulation model at 09:30 UTC and Figure 2d shows predicted stress-equivalent wind vectors at 10 m above the sea surface ($u_{10}$) at 2 km ground resolution from the hourly MeteoFrance AROME operational wind forecast model at 09:00 UTC. The OSCAR L2 swath can be seen outlined in black, overlain on both model outputs.

Overall, very good visual agreement can be seen between OSCAR and the two models, both in the magnitude and direction of both TSCV and OSVW.  OSCAR features additional complexity in the observed current field than the model, with sharper gradients and a separate sub-mesoscale filament close to land in the northern jet north of the island. To the south of the domain, the counter-rotating eddy predicted by the model data (~48.38°N) is also captured in the OSCAR TSCV, albeit with subtle spatial differences and lower magnitude in the observations. Importantly, OSCAR reveals tidal currents north of the island that are both more intense and further from land (due to leeward sheltering of the tidal flow). For OSVW, good similarity can be seen between the retrieved OSVW from OSCAR (Fig. 2b) and AROME model data (Fig. 2d), with fine scale variations in the near-surface winds present in the OSCAR data, especially around the western tips of the island which could originate from

local orographic perturbation. The large-scale north-south gradient in wind speed present in the numerical model is also observed in the OSCAR data, albeit with much finer-scale detail due to the extreme difference in resolution.

Further evidence of the exceptional accuracy of OSCAR to correctly measure complex hydrodynamic structures around Ushant is shown in Figure 3 via comparison and validation against surface current vectors derived from the X-band radar on La Jument lighthouse. Figure 3a shows surface currents derived from 10 minutes of sequential X-band intensity imagery acquired over a radius of 3.1 km around the lighthouse at 09:30 UTC on the 17[th] May, shown with a pixel ground resolution of 200 m, a vector posting of 400 m and the coincident OSCAR L2 swath overlain. Figure 3b shows co-located and superimposed vectors from

both sensors plotted at 400 m posting. The two instruments show very good agreement, with a root mean square difference (RMSD) between the data of 0.18 m s$^{-1}$ for current velocity and 5.27° for current direction (Figure 3c). Both the OSCAR system and the X-band radar measure the southern tidal jet further west than appears in the model (Fig. 2c) and a northward shift in the counter-rotating flow. This is reflected by the RMSD between MARS2D and the measurements from these two remote sensing systems, which report an RMSD of 0.61 m s$^{-1}$ and 6.5° between OSCAR and the model, and 0.47 m s$^{-1}$ and

16.6° between the X-band and the model for current velocity and current direction respectively. The high level of agreement between the two independent remote sensing measurements demonstrates the value of these systems to reveal and resolve inaccuracies that can occur in models of high energy coastal environments.

More detailed examination of Figure 3b Indicates that the best agreement can be seen at the western edge of the OSCAR swath,

which corresponds to higher incidence angles. The comparison around the position of the high tidal flow (at lower incidence angles) is better for TSCV magnitude than it is for direction, with a clear shift in measured current direction between OSCAR and the X-band radar data. The validation of TSCV components from OSCAR and the X-band radar (Figure 3c) reveals good agreement in U (east component; $R^2 = 0.7$) and excellent agreement in V (north component; $R^2 = 0.96$) across a wide range of flow conditions. The combined RMSD between the two sensors for TSCV at 200m ground resolution is found to be 0.19 m s$^{-}$

$^1$. The departure between the two sensors in negative component velocities is related to the different OSCAR TSCV directions observed in the southern tidal jet.

Further validation of the ability of OSCAR to accurately measure fine-scale hydrodynamic structures in such a dynamic coastal environment is seen in Figure 4. Here OSCAR TSCV vectors from 09:38 UTC on the 17[th] May are overlain over a geo-

registered SAR image from the NovaSAR-1 satellite around Ushant. The satellite image was acquired at 10:30 UTC, less than an hour after the OSCAR track, but still within the same ebb-tidal flow regime. The NovaSAR-1 S-band $\sigma^0$ image displays variations in Normalised Radar Cross Section ($\sigma^0$) linked to the modulation of the ocean surface roughness by strong horizontal shear in the flow, producing clear intensity gradients that are coincident with these hydrodynamic structures. Excellent spatial correlation can be observed between the OSCAR TSCV vectors and these gradients in the SAR image, with good

correspondence in the positions of both northern and southern tidal jets around Ushant in the spaceborne SAR $\sigma^0$ and OSCAR

data. The NovaSAR-1 image also displays lighter gradients that coincide with both the counter-rotating flow to the south-west of Ushant and the small flow 'filament' that are observed in the OSCAR data but are not present in the MARS2D output. The comparison with NovaSAR-1 confirms the validity of OSCAR TSCV placing the intense horizontal shear of the ebb-tidal jet north of the island (~48.48°N) approximately 1 km northward of the position predicted with MARS2D (Fig. 2c). The same is true for the position of the southern tidal jet towards the lower extremity of the OSCAR swath. This is an important result, as the exact position of such strong gradients in tidal flows such as these can be challenging to accurately predict using numerical models.

## 4.2 OSCAR results for flood tide on 22 May 2022

Figure 5 shows TSCV and OSVW results from an OSCAR acquisition flying north (looking left) covering part of the flood-tidal regime around Ushant on the $22^{nd}$ May, 2022 at 05:48 UTC. Sea state conditions at the time of acquisition consisted of a light swell from the south-west with a significant wave height of 1 m and a peak period of 9 s. The wind was generally from the north-east around 5-6 m.s$^{-1}$. Figure 5a and Figure 5b show OSCAR retrieved TSCV and OSVW respectively. Figure 5c shows MARS2D simulated currents around Ushant at 05:45 UTC and Figure 5d shows forecast $u_{10}$ wind vectors at 06:00 UTC from the AROME model. Once again, OSCAR present very good overall agreement with the two models, both in the magnitude and direction of both TSCV and OSVW. Moreover, the results confirm the ability of OSCAR to resolve not only high tidal currents but also current fields in quiescent areas, for example in the shallow coastal areas around western Ushant that are sheltered from the northward flood tide which is present in both the predicted and observed TSCV fields. Similar to the ebb-tide case on the $17^{th}$ May (previous section), the OSCAR TSCV field shows a more northerly extent of the accelerated tidal flow within the imaged swath than in the predicted data; highlighting the stark differences that can be obtained between observations and numerical models in such dynamic coastal areas. The OSCAR OSVW (Fig. 5b) captures the same general north-south trend and variability seen in the forecast AROME wind data (Fig. 5d), with the reduction in wind speeds to the lee of the island clearly visible in both the OSCAR results and the predictions.

On the $22^{nd}$ May, three OSCAR acquisitions were made over the same area to the west of Ushant within a period of 15 minutes, with two passes in a southerly direction and one travelling north. The objective was to demonstrate the consistency of the OSCAR L2 products between successive passes and in different orientations. Figure 6 shows the comparison of TSCV for the three tracks (shown as coloured vectors) co-located on the same grid and overlain on MARS2D predicted currents (black vectors). The swath outlines are in orange for the northward track at 05:48 UTC and in blue for the southward tracks at 05:39 UTC and 05:54 UTC (for clarity only one swath is outlined for the two southwards tracks). The agreement between successive passes is excellent, with different TSCV vectors nearly always perfectly superposed and quasi-undistinguishable at most grid points. The TSCV median sample standard deviation between the three flights (with each TSCV vector treated as an independent measurement, i.e., n = 3) is 0.10 m s$^{-1}$ for current velocity and 1.18° for current direction. It is worth noting that the radar lines-of-sight relative to the wind were not particularly favourable in this instance. As the wind direction was from

the north-east (Fig. 5d), the antenna squint angles 45° and the flight directions either north or south, the southward flights at 05:39 and 05:54 UTC (blue and green) correspond to one squinted beam looking crosswind and the other looking up-wind, whereas the northward flight at 05:48 UTC (orange) had one squinted beam looking crosswind and the other downwind. This is important, as the performance of the three-look concept is known to be impacted when one of the azimuth lines-of-sight is aligned with the wind (Stoffelen and Portabella, 2006; Martin et al., 2018). However, the consistency between the TSCV

obtained with up-wind (blue and green) and down-wind (orange) lines-of-sight is seen to be very good at those points where the swaths from the northward and southward flights overlap.

## 5 Discussion, conclusions and future work

This paper presents the results of the first scientific airborne campaign of the OSCAR high-squint, three-look SAR-ATI instrument and represents the first demonstration of simultaneous retrieval of TSCV and OSVW in a single pass using a

SeaSTAR-type observing concept in a highly dynamic coastal marine environment. The results are extremely promising, with OSCAR and the SeaSTAR inversion algorithm reporting a high degree of accuracy and self-consistency over a large swath in highly inhomogeneous coastal conditions. The OSCAR instrument demonstrates the capability to provide quantitative information about fine scale dynamics that are not correctly represented with numerical models and are difficult to observe with ground-based sensors alone. In situ observations of hydrodynamics in high energy, macrotidal environments are costly

and challenging (Neil and Hashemi, 2018) and provide only limited spatial detail needed to understand frontal dynamics in areas of strong spatial gradients. HF radars could provide synoptic information about ocean currents but their coverage is patchy and limited to a few coastal regions in some industrialised countries. Whilst HF radar data was recorded during the campaign, coverage around the island of Ushant, the focus of this paper, is unfortunately lacking; especially in the area to the West of the island that sits in the radio shadow of the HF ground stations on the French mainland. HF coverage of the open-

ocean homogeneous site sampled in the campaign was good, however, and their analysis and comparison with OSCAR will be a subject of future work.  The quality of the OSCAR data obtained from these four flights spanning two days is notably impressive. Comparisons of the three flights for the 22[nd] May, acquired within a short time over roughly the same ground track but with radically different radar viewing directions with respect to the wind, give confidence in the OSCAR and SeaSTAR current and wind vector retrieval. Comparisons with high-resolution data from X-band radar, NovaSAR-1 and models

demonstrate that the OSCAR airborne instrument and the SeaSTAR mission concept clearly bring great additional capability to the research community and present great promise for increasing our understanding of small-scale processes in coastal and shelf seas and marginal ice zones.

It is noteworthy that the retrieval returns excellent results within this exceptionally heterogeneous coastal environment whilst

using the open-ocean wind-dependent WASV correction by Mouche et al. (2012). Importantly, no specific adjustments were made to account for additional sea state dependency of the WASV on factors like fetch, wave age, wave breaking at fronts or

wave-current interactions over shallow bathymetry. Our results align with prior suggestions from Martin et al (2017) that indicate that the effects on SAR-ATI of strong current gradients and wave age may hold secondary importance in these highly dynamic areas. There is a common perception that the accuracy of available open-ocean GMFs may be insufficient to predict

the WASV in areas of strong surface current gradients, and may require mixed-polarisation sensor capability to correctly address additional Doppler effects at fronts (Kudryavstev et al., 2014; Martin et al., 2018). A surprising result of our work is the consistency of TSCV retrieved using the C-DOP GMF of Mouche et al. (2012) and its predicted WASV in such a dynamic coastal tidal environment. The C-DOP GMF captures the average behaviour of the WASV based on a large globally distributed dataset of Envisat ASAR Doppler measurements over the open ocean (i.e., hydrodynamically homogenous, microtidal

environments) and as such does not contain the WASV response to shallow water dynamics, wave-current interactions, strong tidal shear, etc. Additionally, the C-DOP GMF has been adapted for this work in both frequency (from C-band to Ku-Band use) and extended in its upper applicable range of incidence angle from 44° to 62°. Initial comparison tests applying the GMF of Yurovsky et al (2019) showed minor differences with our extended version of C-DOP at these high incidence angles. A more in-depth study of the WASV and the effect of these different GMFs will be made in a future study. The quality of the

OSCAR retrieved TSCV in the complicated Iroise Sea environment suggests that the assumed sensitivities of the WASV to hydrodynamic processes are perhaps not as important as previously assumed. Future work will consider other GMFs to compute the WASV that include dependence on ocean wave parameters such as has been developed by Yurovsky et al. (2019). TSCV retrieval performance is especially sensitive to the choice of WASV at low incidence angles (Martin et al, 2018), so a different GMF may further improve the OSCAR results in the near-range parts of the swath. The full wind and waves

parametrised model for Yurovsky et al (2019) has not been tested with OSCAR data at this time but will be the basis of a larger validation study in the future.

All results of this study were obtained using the simultaneous retrieval of current and wind vectors that is the chosen baseline algorithm for OSCAR and SeaSTAR Level-2 inversion. But other inversion approaches exist, for example one can consider a

sequential retrieval approach where ancillary wind information (for example from AROME) serves to correct the WASV directly in each line-of-sight, before recovering the current vector (as in Martin et al., 2017). In this instance, computation time is almost instantaneous, versus a typical ~30 minutes computation on a 20-core machine to apply simultaneous inversion to a single track at 200m resolution. However, our tests (not shown) indicate that the quality of the TSCV is degraded when using sequential approach. The degradation is visible in the poorer performance against X-band marine radar currents, but also in

the mis-localisation of OSCAR retrieved current gradients with the visible roughness gradients in NovaSAR-1 imagery. This is particularly noticeable at low incidence angles and may be linked to errors inherent to the WASV GMF mentioned previously.

TSCV is defined as the effective mass transport, being composed of any movement of the sea surface that is not accounted for

by the WASV, i.e. from the wind (in the case of the GMF of Mouche et al, 2012; or wind and waves in the case of Yurovsky

et al, 2019). In this sense, TSCV contains contributions from a wide variety of physical processes that may be isolated depending on the analysis or secondary information used. In the case of the work presented in this paper the TSCV is completely dominated by the contribution from the tide, being orders of magnitude greater than, e.g., Stokes drift or Ekman currents. Being a surface imaging system, OSCAR and the SeaSTAR methodology are also capturing only the ocean motion that is manifest at the surface. The currents sensed by the X-band radar used for comparison are barotropic, as are the currents produced by the MARS2D numerical model (which is depth-averaged). Unravelling baroclinic contributions to the Doppler signal sensed at the surface will require a different experimental design in an area that is not dominated by tidal currents (e.g., the open ocean).

The comparison between OSCAR and ground-based X-band radar data is excellent, especially when considering the differences in frequency and imaging mechanisms between the two systems. Some consideration should be given to these differences and how they may account for a proportion of the scatter seen in Figure 3c. For the X-band radar (e.g., using the methods of Streßer et al., 2017), the resulting derived currents represent a weighted mean over the upper ocean, where the greatest weight is assigned to the surface and the effective depth of the current relates to the maximum ocean wavelengths imaged by the radar and considered in the current fit. For OSCAR, the TSCV derives from Doppler shifts in backscatter at moderate, non-grazing, angles (incidence angle as steep as 22°) where there are combined effects from small-scale surface roughness and longer ocean wavelengths, particularly wind waves around 10 m (Chapron et al., 2005). The agreement between OSCAR and the X-band derived surface currents in Figure 3b is seen to be better in one area (the Northerly vectors in the centre-left of the swath) than others (the bottom-right of the swath). Aside from differences in imaging and retrieval methods between the two data, another source of the difference may be the combination of antenna look direction relative to the surface current and errors in the WASV correction at low incidence angles in the near range of the swath. At these low incidence angles the WASV is more sensitive to the wind (Mouche et al, 2012) which in the simultaneous retrieval also has its own sources of error, e.g., from the calibration of NRCS and its effect on the output of the NSCAT-4DS GMF. Combined with the fact that the current is flowing in broadly the same direction as one of the look azimuths (for the aft beam) and perpendicular to another (the fore beam), errors in the WASV could be magnified more than in other areas of the swath. This is a complicated interplay of calibration factors, SAR interferometry and GMFs that warrants further investigation. Further investigations are also needed to better understand differences in TSCV measured by different sensors, including also HF radars and other ocean current sensors like ADCP, and how to accommodate these differences when validating new sensors like OSCAR.

This paper presents only a preliminary validation of the OSCAR data to demonstrate the innovative and promising spatial mapping capabilities of this new system in the extreme macrotidal coastal environment close to Ushant. Fuller validation of the OSCAR L2 results is necessary however to quantitatively compare the OSCAR data against established ocean measurements from ADCP and HF radar also collected during the SEASTARex project. For this, the focus of the validation must shift south, to a second instrumented site set up during SEASTAREx to validate OSCAR in a geophysically homogenous

site to the south of Ushant (centred on 48.256°N, 5.249°W). Those larger datasets and comparisons will be the object of a separate publication. More recently still, OSCAR flew in its second scientific campaign in May 2023 in the north-west Mediterranean Sea (north of the island of Menorca, Spain) famous for strong sub-mesoscale ocean dynamics. The OSCAR-Med airborne flights were timed to coincide with overpasses of the Surface Water and Ocean Topography (SWOT) mission during its 1-day fast-repeat Cal/Val phase, and in situ measurements from the BioSWOT-Med oceanographic ship campaign

(Doglioli and Gregori, 2023). The results of the OSCAR-Med campaign will be the subject of a future study. Finally, given the scientific objectives of the SeaSTAR mission concept to measure TSCV and OSVW in marginal ice zones, the team is currently exploring opportunities to fly a new OSCAR scientific campaign close to the sea ice edge.

**Code availability**

The repository for the SeaSTAR processing software used for the work in this study is available at

https://doi.org/10.5281/zenodo.10026593

**Data availability**

OSCAR data from the Iroise Sea campaign is available from ESA's Earth Online database (doi: 10.57780/esa-633ce94) and can also be provided by the corresponding author upon request.

**Author contribution**

AM, CG, JH, LM, JM, MP and AMe planned the campaign, TC and PM-I contributed to the campaign planning and realization, AMe managed the recording of the airborne data, DM, AM, KM and RC developed the processing software, KM, JM and MP performed the OSCAR level-1 data processing and calibration, AM and DM performed the OSCAR level-2 data processing, RC, JH and LM recorded and processed the ground truth data, DM wrote the manuscript, RC and JH provided text input for the manuscript, DM, AM and CG reviewed and edited the manuscript.

**Competing interests**

The authors declare that they have no conflict of interest.

**Acknowledgements**

The authors would like to thank Christian Trampuz (formerly of Metasensing BV), Hugo Kerhuel and Thomas Jamne for the preparation and execution of the OSCAR Iroise Sea campaign in May 2022. The authors would also like to thank: Nicholas

Gebert from ESA who was the initial Technical Officer for the OSCAR system development, Rui Duarte and team at France Energie Marine (FEM) who provided essential access to La Jument lighthouse and support for the installation and operation of the X-band radar, Thiago Luiz at Metasensing BV for his work in the SAR processing of OSCAR data and Guisseppe Greco at ICM-CSIC, Spain and Wenming Lin at NUIST, China for their help with the radiometric calibration of OSCAR data. Many thanks to Clive Neil at NOC for his essential assistance in setting up the Seastar project software repository.

**Financial support**

This study was supported by the SEASTARex contract from the European Space Agency (400017623/22/NL/IA). The OSCAR instrument was developed by Metasensing BV under funding from the European Space Agency in the OSCAR contract (4000116401/16/NL/BJ).

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

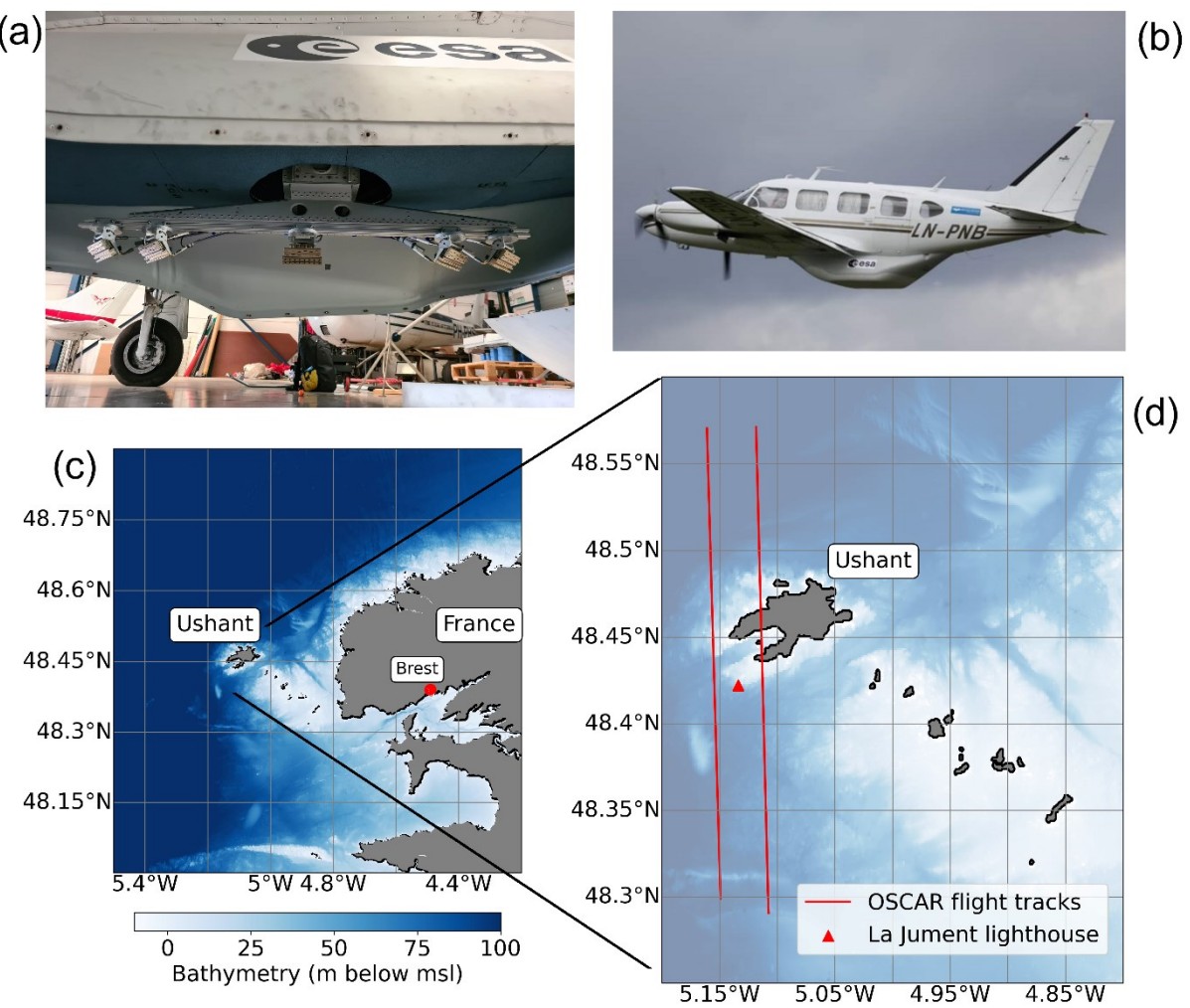

**Figure 1: (a) The OSCAR 3-look Ku-Band SAR instrument and 3-axis stabilisation gimbal; the two fore and aft ATI antenna pairs can be seen to the left and right of the boom with the scatterometer mid antenna in the centre. (b) The OSCAR instrument within the radome pod mounted to MetaSensing's Piper Navajo airframe. (c) The study area of the Iroise Sea, France, bathymetry in metres below mean sea level from EMODnet. (d) The island of Ushant with positions of La Jument lighthouse and the over-island flight tracks marked.**

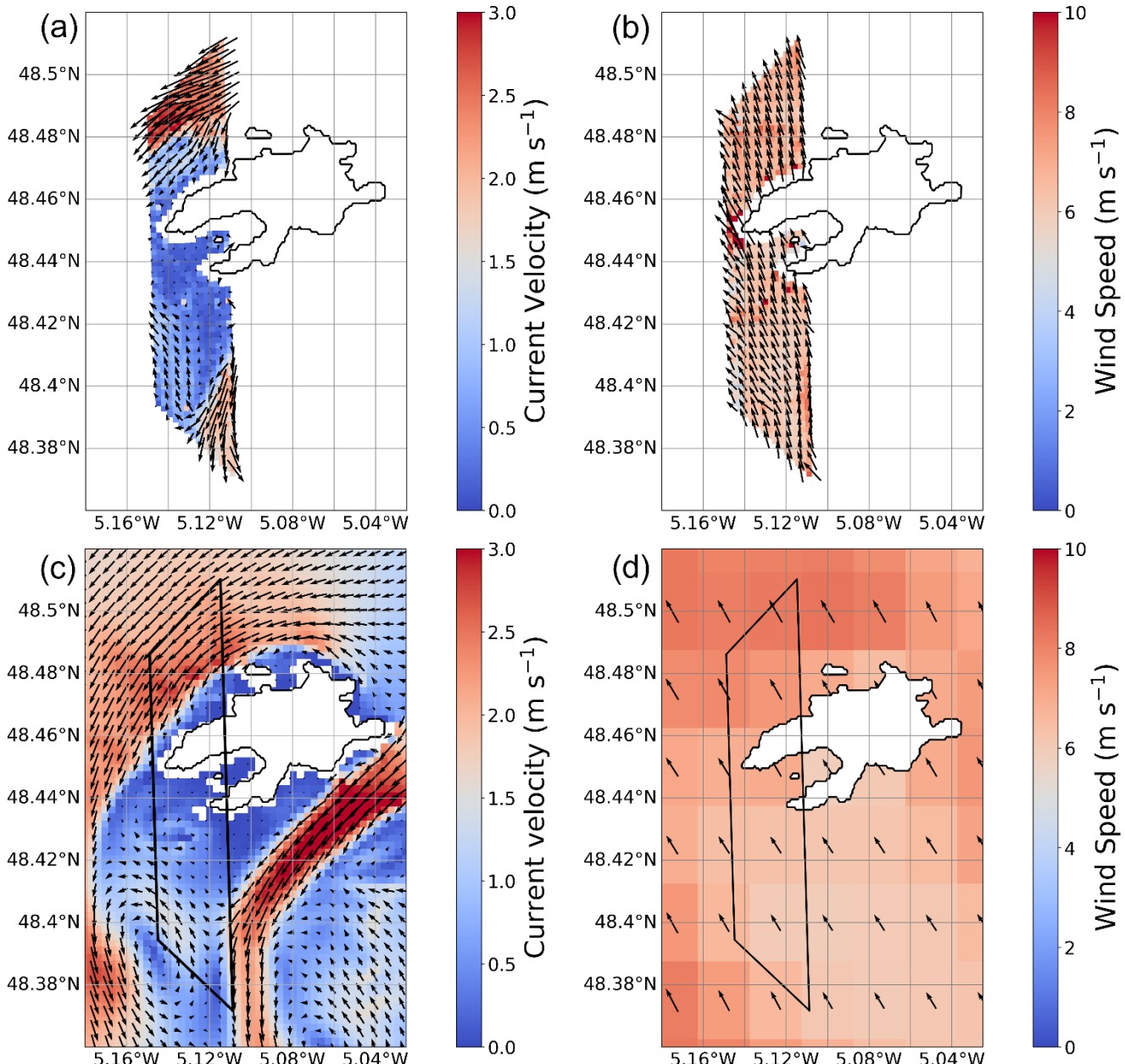

**Figure 2: Simultaneously-retrieved L2 data products from a single OSCAR acquisition over Ushant at 09:38 UTC on 17th May 2022 during an ebb tide: (a) OSCAR total surface current vector, (b) OSCAR ocean surface vector wind, (c) depth-average current velocity from the MARS2D model at 09:30 UTC and (d) Wind speed ($u_{10}$) from the AROME operational wind model at 09:00 UTC. The outline of the OSCAR L2 swaths are shown in black on the two model outputs.**


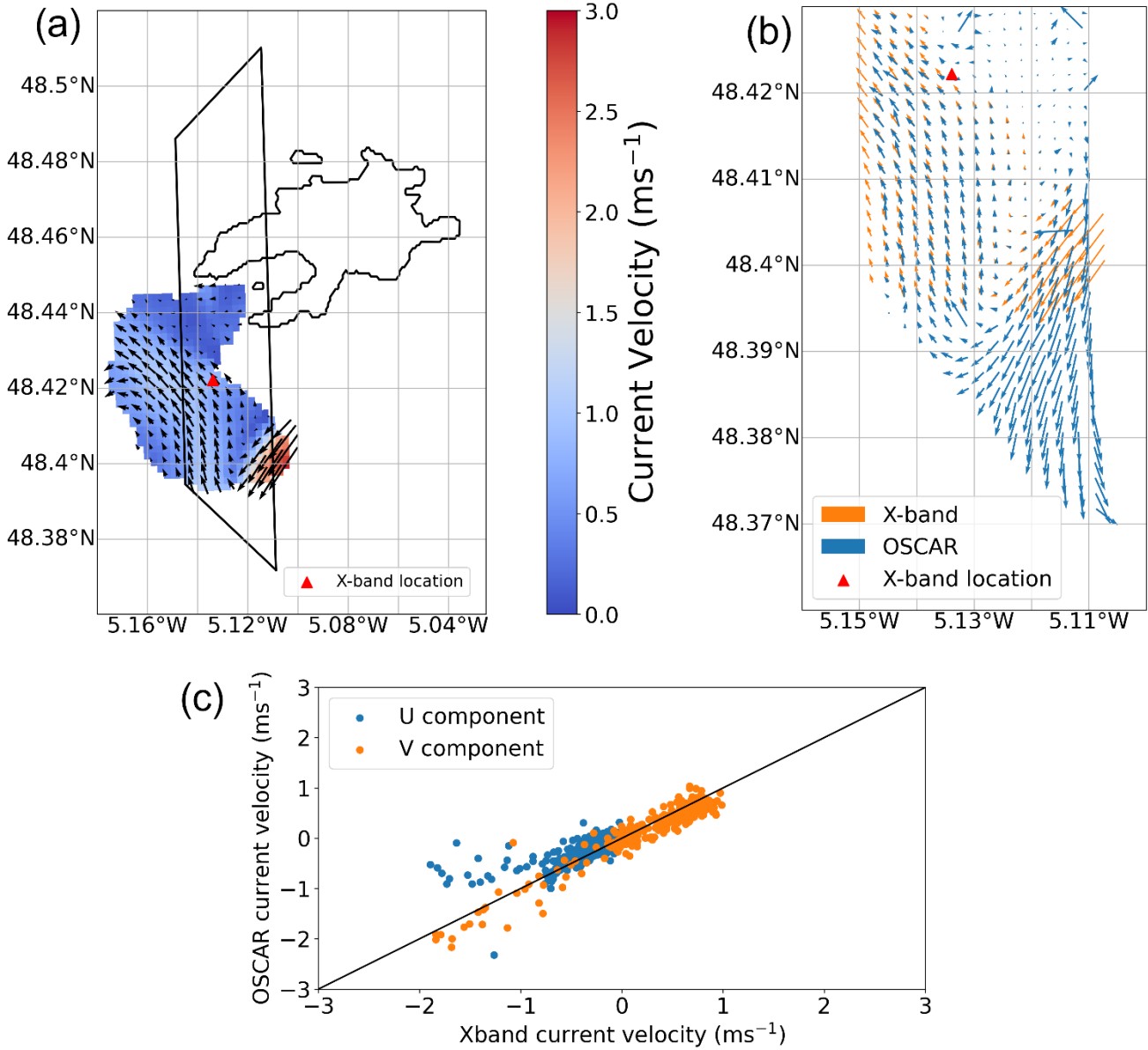

**Figure 3: (a) X-band derived surface current vectors from La Jument lighthouse from 10 minutes of radar data at 09:30 UTC on**
**the 17th May 2022, (b) Co-located OSCAR total surface current vectors at 09:38 UTC and X-band current vectors, (c) U (east) and**
**V (north) current component direct comparison between the two sensors.**


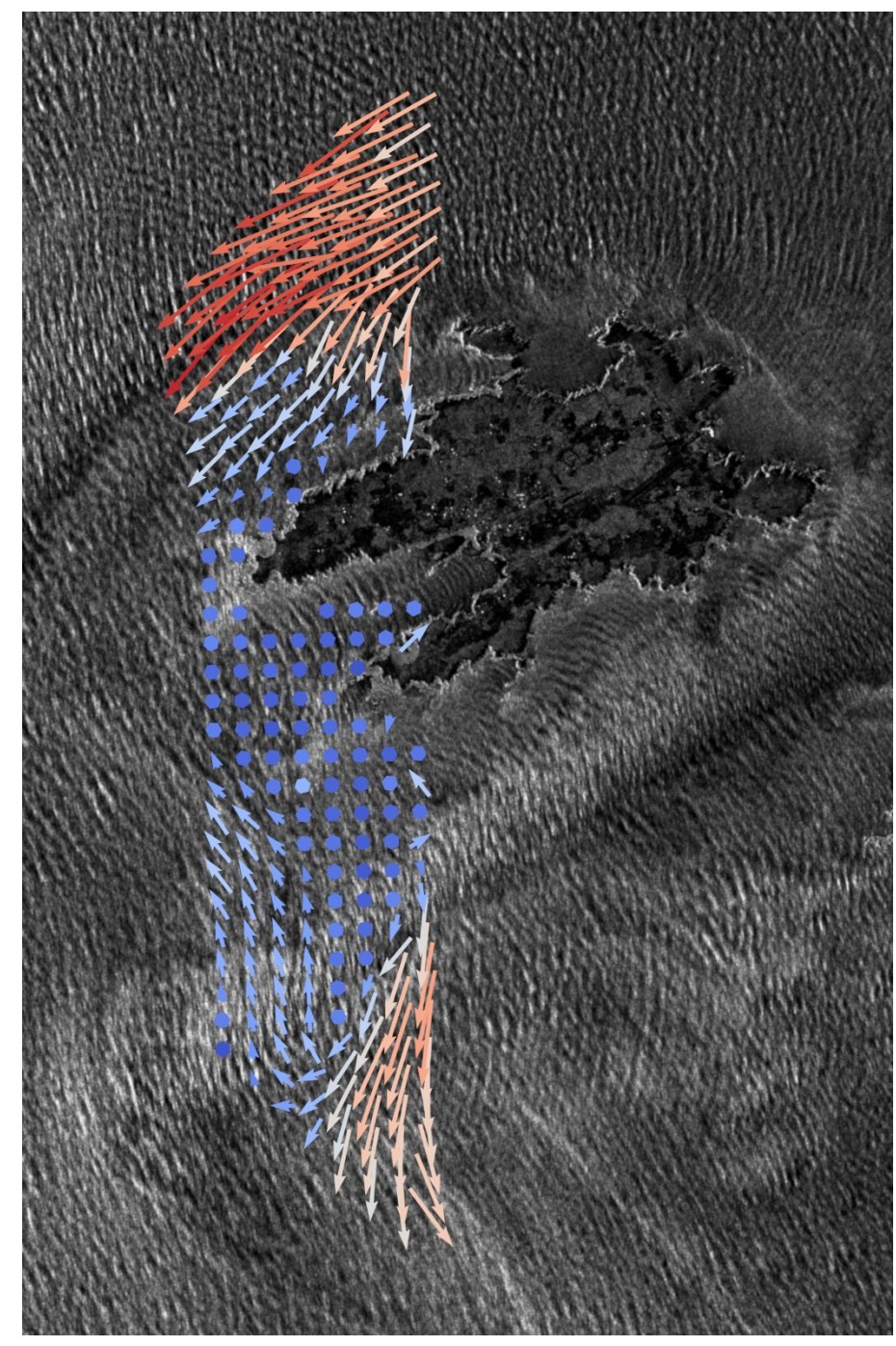

**Figure 4. OSCAR total surface current vectors from 09:38 UTC on the 17th May 2022, overlain on a NovaSAR-1 S-band $\sigma^0$ image acquired at 10:30 UTC of Ushant and the surrounding waters during a period of ebb-tidal flow. NovaSAR-1 image courtesy of SSTL and Airbus.**


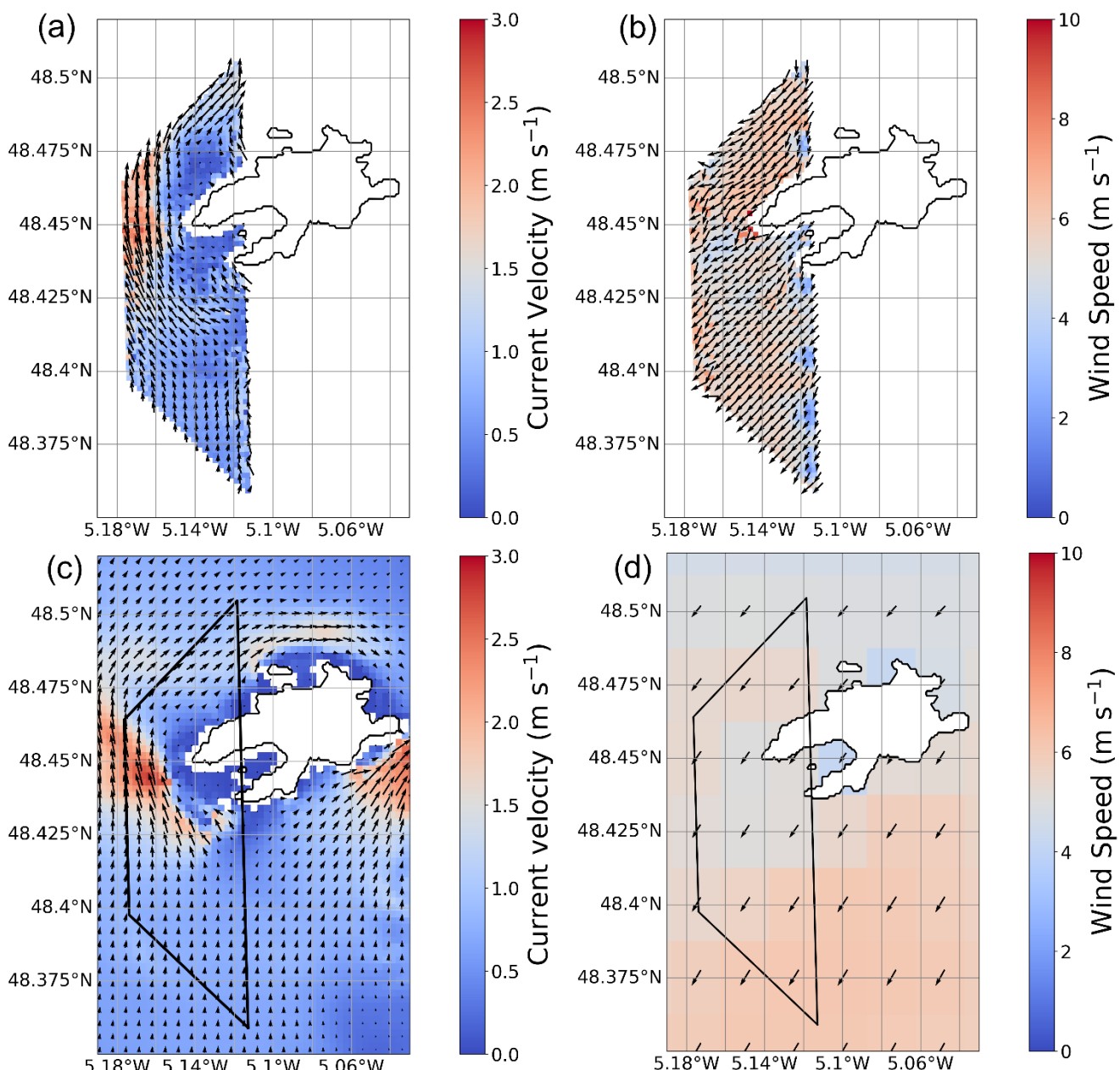

**Figure 5: Simultaneously-retrieved currents and winds from a single OSCAR acquisition over Ushant at 05:48 UTC on 22nd May 2022 during a flood tide: (a) OSCAR total surface current vector, (b) OSCAR ocean surface vector wind, (c) depth-average current velocity from the MARS2D model at 05:45 UTC and (d) Wind speed (u10) from the AROME operational wind model at 0600 UTC. The outline of the OSCAR L2 swaths are shown in black on the two model outputs.**

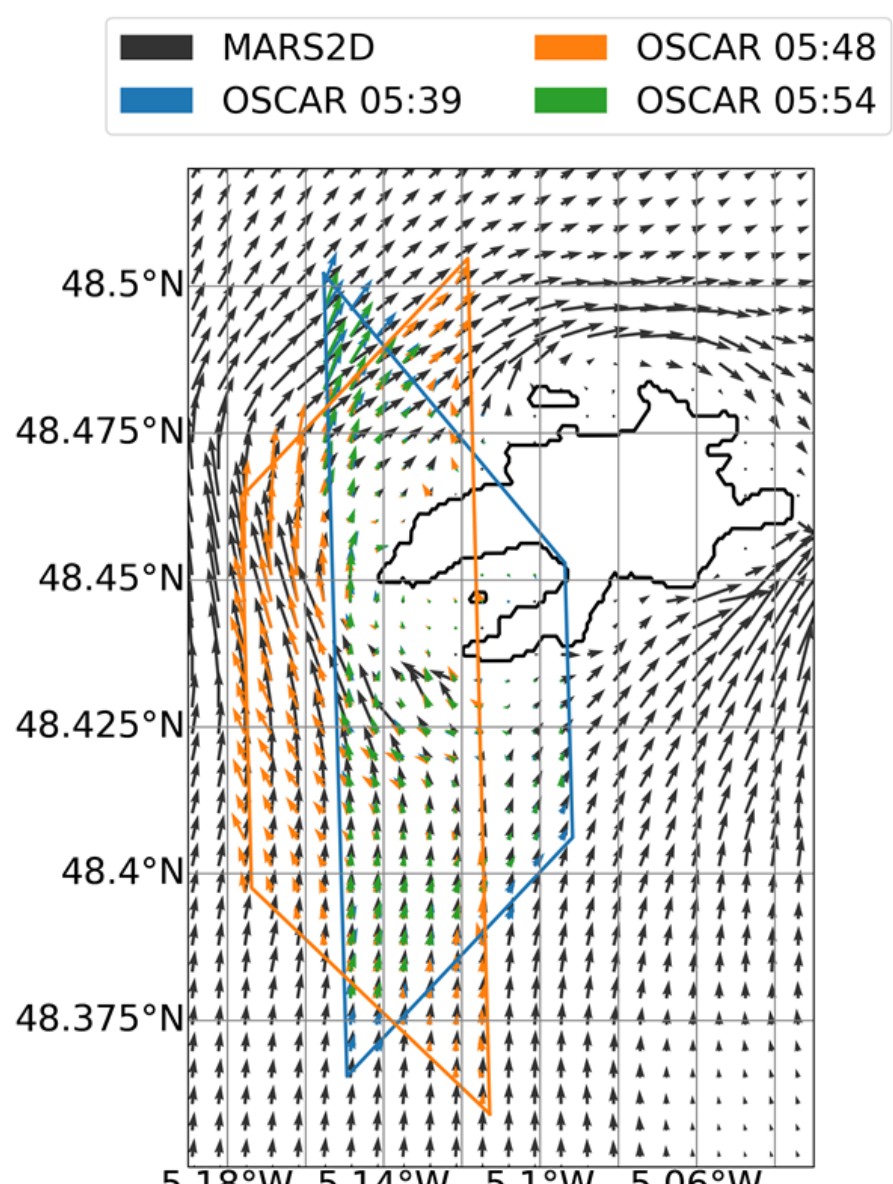

**Figure 6: Total surface current vector comparison between three successive OSCAR tracks on the 22nd May, co-located onto a common grid and overlain on MARS2D current vectors from 05:30 UTC (black arrows). OSCAR swath outlines are depicted in their corresponding vector colour (blue, 05:39 UTC; orange, 05:48 UTC; green, 05:54 UTC) . The tracks from 05:39 UTC and 05:54 UTC sharing the same swath outline.**