# Peer review of "A new airborne system for simultaneous high-resolution ocean vector current and wind mapping: first demonstration of the SeaSTAR mission concept in the macrotidal Iroise Sea"

_EGUsphere, 2023_

## Referee Comment (RC1)

**A new airborne system for simultaneous high-resolution ocean vector current and wind mapping: first demonstration of the SeaSTAR mission concept in the macrotidal Iroise Sea**

**General comments**

This manuscript presents the very first validation from real observations of the concept of squinted 3-look SAR along-track interferometry to retrieve the vectors of surface wind and total surface current velocity mapped at high resolution (200m).

The observations were collected with an airborne SAR system called OSCAR in a coastal environment characterized with high tidal flow. OSCAR is the airborne demonstrator of the Seastar satellite concept funded by ESA and based on the same principle of along-track interferometry from a 3-look squinted SAR.

Although based on a limited number of cases studied (two meteo-oceano situations, with one sampled under different flight geometries), the results are very convincing. The retrieved field of wind and surface current are validated against current fields from HF radar observations and from numerical model outputs.

So, the manuscript represents an important step in the assessment of this novel concept.

The manuscript is well organized and well written. Overall, it is a very good paper.

**Specific comments**

1- Retrieving the Total Surface Current Velocity is the main goal of the OSCAR and Seastar concepts. However, in the manuscript, the notion of TSCV is not defined: in particular does it include a motion component due to waves, in additional to what is classically defined as "current"? This should be clarified.

If it includes a component due to the waves, then in the validation, the comparison with surface currents from a numerical oceanographic model may not be fully appropriate because this latter does not include the wave component.

So more specifically, I suggest that the authors add some words on the definition on TSCV at the beginning of the manuscript (probably in the introduction) and that they add some discussion in the section 4 and/or 5 on the fact that (probably) wave effects are omitted in the current field from the numerical model whereas it is included in the retrieved TSCV.

2- The cost function defined to retrieved both the surface wind vector and the TSCV (Eq.1) is slightly different from the one presented in Martin et al, 2018 (which I copy below) . Indeed in this latter the GMF for $\sigma_0$ and for the Doppler anomaly is defined as a function of the wind relative to the current ($u_{10}$-c ), whereas in the submitted manuscript, the current effect is not taken into account in the wind or current GMF model functions.

From Martin et al, 2018

$$J_{pp}(\mathbf{u_{10}}, \mathbf{c}) = \frac{1}{2N} \sum_{i=1}^{N} \left( \frac{\sigma_{meas,i}^{0,pp} - KuMod(\mathbf{u_{10}} - \mathbf{c}, pp)}{\Delta\sigma^0} \right)^2$$

$$+ \frac{1}{2N} \sum_{i=1}^{N} \left( \frac{df_{meas,i}^{pp} - KuDop(\mathbf{u_{10}} - \mathbf{c}, pp) + 2. \, c_{//_i}. \sin\theta/\lambda_e)}{\Delta df} \right)^2 \quad (1)$$

From the submitted manuscript

$$J(u_{10}, c) = \frac{1}{N_S + N_D} \sum_{i=1}^{N_S} \left( \frac{KuMod(u_{10}, \chi_i, \theta_i, p_i) - \sigma_{obs,i}^0}{\Delta\sigma_i^0} \right)^2 + \frac{1}{N_S + N_D} \sum_{i=1}^{N_D} \left( \frac{KuDop(u_{10}, \chi_i, \theta_i, p_i) + c_{||_i} - RSV_{obs,i}}{\Delta RSV_i} \right)^2 \quad (1)$$

I suggest the authors explain and justify this evolution or comment on this intertwined relation between wind and current in the GMFs.

3- The empirical model (GMF) used to express the Doppler anomaly due to the wave effects (WASV) is derived from the work of Mouche et al (2012). However, in the paper of Mouche et al, it seems that the model is limited to incidence angles less than about 40°, whereas the observations of OSCAR extend up to about 69° . So, for the inversion of OSCAR data, how is the GMF for the Doppler anomaly extended to the largest incidence angles (from 40 to 69°) ? This should be discussed.

---

## Author Response (AR1)

We kindly thank the reviewers for their comments and suggestions. As follows is a point-by-point response, including associated changes made to the manuscript. **The original reviewer's comments are in black**, our responses are in red and changes to the manuscript are in blue

**Reviewer #1 Specific comments**

1. Retrieving the Total Surface Current Velocity is the main goal of the OSCAR and Seastar concepts. However, in the manuscript, the notion of TSCV is not defined: in particular does it include a motion component due to waves, in additional to what is classically defined as "current"? This should be clarified.

If it includes a component due to the waves, then in the validation, the comparison with surface currents from a numerical oceanographic model may not be fully appropriate because this latter does not include the wave component. So more specifically, I suggest that the authors add some words on the definition on TSCV at the beginning of the manuscript (probably in the introduction) and that they add some discussion in the section 4 and/or 5 on the fact that (probably) wave effects are omitted in the current field from the numerical model whereas it is included in the retrieved TSCV.

Yes this is the case - throughout a majority of the comparison field (i.e., imaged area with X-band crossover) the vectors are broadly North-South orientated so the V velocity component shows a better comparison. We felt that along with figure 3b that a vector component comparison would be a better way of showing where our results are good and where they are less good - specifically in the high flow region that happens to coincide with low incidence angles. We will add some more discussion of this in Section 5.

(Line 64:67) The total surface current vector (TSCV), corresponding to the effective mass transport, contains contributions from surface wind drift, Ekman currents, geostrophic currents and Stokes drift but is dominated by tidal currents in many coastal environments by an order of magnitude. TSCV is derived from the surface motion directly measured by the Doppler shift minus the WASV.

(Line 369:378) TSCV is defined as the effective mass transport, being composed of any movement of the sea surface that is not accounted for by the WASV, i.e. from the wind (in the case of the GMF of Mouche et al, 2012; or wind and waves in the case of Yurovsky et al, 2019). In this sense, TSCV contains contributions from a wide variety of physical processes that may be isolated depending on the analysis or secondary information used. In the case of the work presented in this paper the TSCV is completely dominated by the contribution from the tide, being orders of magnitude greater than, e.g., Stokes drift or Ekman currents. Being a surface imaging system, OSCAR and the SeaSTAR methodology are also capturing only the ocean motion that is manifest at the surface. The currents sensed by the X-band radar used for comparison are barotropic, as are the currents produced by the MARS2D numerical model (which is depth-averaged). Unravelling baroclinic contributions to the Doppler signal sensed at the surface will require a different experimental design in an area that is not dominated by tidal currents (e.g., the open ocean).

2. The cost function defined to retrieved both the surface wind vector and the TSCV (Eq.1) is slightly different from the one presented in Martin et al, 2018 (which I copy below) . Indeed in this latter the GMF for s0 and for the Doppler anomaly is defined as a function of the wind relative to the current (u10-c ), whereas in the submitted manuscript, the current effect is not taken into account in the wind or current GMF model functions.

From Martin et al, 2018

$$J_{pp}(\boldsymbol{u_{10}}, \mathbf{c}) = \frac{1}{2N} \sum_{i=1}^{N} \left( \frac{\sigma_{meas,i}^{0,pp} - KuMod(\boldsymbol{u_{10}} - \mathbf{c}, pp)}{\Delta\sigma^0} \right)^2$$
$$+ \frac{1}{2N} \sum_{i=1}^{N} \left( \frac{df_{meas,i}^{pp} - KuDop(\boldsymbol{u_{10}} - \mathbf{c}, pp) + 2.\, c_{//i}.\, \sin\theta/\lambda_e)}{\Delta df} \right)^2 \quad (1)$$

From the submitted manuscript

$$J(u_{10}, c) = \frac{1}{N_S + N_D} \sum_{i=1}^{N_S} \left( \frac{KuMod(u_{10}, \chi_i, \theta_i, p_i) - \sigma_{obs,i}^0}{\Delta\sigma_i^0} \right)^2 + \frac{1}{N_S + N_D} \sum_{i=1}^{N_D} \left( \frac{KuDop(u_{10}, \chi_i, \theta_i, p_i) + c_{||i} - RSV_{obs,i}}{\Delta RSV_i} \right)^2 \quad (1)$$

I suggest the authors explain and justify this evolution or comment on this intertwined relation between wind and current in the GMFs.

In Martin et al., 2018, u10 was considered as being the Earth Relative Wind (same as for Normalised Wind Product, NWP). Due to confusion in the community, where u10 is considered as the Ocean Surface Wind (Earth Relative Wind minus Current), we used this later definition in the submitted manuscript. Thank you for raising it, we will highlight it and make reference to the difference with Martin et al., 2018.

 (Line 113:118) It should be noted that aside from the adaption for three simultaneous looks, Equation 1 differs from Martin et al (2018), specifically in the definition of the wind vector. In Martin et al (2018), $u_{10} = u_{ERW}$ is defined as the Earth relative wind (same as for the Numerical Weather Prediction NWP product), in this paper $u_{10} = u_{OSVW}$ is the stress equivalent Ocean Surface Wind at 10 m height which is equivalent to the difference between the Earth relative wind and the surface current, i.e. $u_{OSVW} = u_{ERW} - c$. Despite $u_{ERW}$ being the wind used in numerical weather prediction, the GMFs for $\sigma^0$ and Doppler shift used are at first order functions of $u_{OSVW}$ and not $u_{ERW}$, hence this slight modification.

3. The empirical model (GMF) used to express the Doppler anomaly due to the wave effects (WASV) is derived from the work of Mouche et al (2012). However, in the paper of Mouche et al, it seems that the model is limited to incidence angles less than about 40°, whereas the observations of OSCAR extend up to about 69° . So, for the inversion of OSCAR data, how is the GMF for the Doppler anomaly extended to the largest incidence angles (from 40 to 69°) ? This should be discussed.

Thank you for raising this point that we overlooked. Indeed Mouche et al., 2012 has only been developed up to 44° of incidence angle. The model implemented in our methodology has been extended to higher incidence angles. From Yurosky et al's 2019 results (with incidence angles up to 65 degrees), this is sensible. This extension to higher incidence angle will be highlighted in the Methodology (Section 2.2)  and will be discussed more fully in Section 5.

(Line 120:126) For the results presented in this paper the $KuDOP$ GMF used for Equation 1 is an adapted version of the Envisat-ASAR C-band (5.4 GHz) empirically derived 'C-DOP' GMF (Mouche et al. 2012). The RSV derived from this model has proven to vary little with radar frequency and to be applicable over a wide range with examples in 9.8 GHz X-band (Martin et al., 2016; Elyouncha et al., 2024) up to 35.7 GHz Ka-band (Yurovsky et al., 2019; Polverari et al., 2022). The range of incidence angle for C-DOP is here extended up to 62° (empirically derived only up to 44°). These assumptions will be discussed in Section 5. Here C-DOP has been adapted in two ways: firstly, for Ku-Band use via a frequency shift of the Doppler calculation and secondly, extending the model's applicable upper range of incidence angle from 44° to 62°.

**Reviewer #2 specific comments**

The paper is very well written and the results show a promising technology to measure ocean surface velocity at very high resolution over relatively large areas in a synoptic way. The only concern that I have is if Ocean Sciences is the correct place to publish this work since it presents a technological development rather than providing insights on ocean processes (however this is an opinion and I agree that this is far away from the role of a referee). The authors claim that more experiments are required to fully demonstrate the capabilities of the system and they prepare a more detailed work with additional tests.

On your comment concerning the fit with OS, we felt that as this manuscript contains the first results of a new system for oceanographic remote sensing that a broader oceanographic audience would be interested rather than restricting these first results to a more specialist remote sensing publication. Previous papers similar to this have been published in OS (e.g., https://os.copernicus.org/articles/16/1399/2020 which gave us confidence that it is indeed a good fit.

Ln 235: Has this better agreement in the y-direction something to do with the preferential direction of the flow?.

Yes this is the case - throughout a majority of the comparison field (i.e., imaged area with X-band crossover) the vectors are broadly North-South orientated so the V velocity component shows a better comparison. We felt that along with figure 3b that a vector component comparison would be a better way of showing where our results are good and where they are less good - specifically in the high flow region that happens to coincide with low incidence angles. We will add some more discussion of this in Section 5.

(Line 387:398) The agreement between OSCAR and the X-band derived surface currents in Figure 3b is seen to be better in one area (the Northerly vectors in the centre-left of the swath) than others (the bottom-right of the swath). Aside from differences in imaging and retrieval methods between the two data, another source of the difference may be the combination of antenna look direction relative to the surface current and errors in the WASV correction at low incidence angles in the near range of the swath. At these low incidence angles the WASV is more sensitive to the wind (Mouche et al, 2012) which in the simultaneous retrieval also has its own sources of error, e.g., from the calibration of NRCS and its effect on the output of the NSCAT-4DS GMF. Combined with the fact that the current is flowing in broadly the same direction as one of the look azimuths (for the aft beam) and perpendicular to

another (the fore beam), errors in the WASV could be magnified more than in other areas of the swath. This is a complicated interplay of calibration factors, SAR interferometry and GMFs that warrants further investigation. Further investigations are also needed to better understand differences in TSCV measured by different sensors, including also HF radars and other ocean current sensors like ADCP, and how to accommodate these differences when validating new sensors like OSCAR.

Please define TSCV

This comment chimes with Reviewer #1, so we will be providing more detail in the Introduction and Section 5, however briefly here:

TSCV corresponds to the effective mass transport. It does include the Stokes drift, but does not include waves artifacts (Wind-wave Artifact Surface Velocity, WASV) in the direct Doppler measurements.

(Line 64:67) The total surface current vector (TSCV), corresponding to the effective mass transport, contains contributions from surface wind drift, Ekman currents, geostrophic currents and Stokes drift but is dominated by tidal currents in many coastal environments by an order of magnitude. TSCV is derived from the surface motion directly measured by the Doppler shift minus the WASV.

Is the measured surface ocean velocity the geostrophic component plus the ekman and stokes components?

We measure the total surface current, which in the case here is dominated by the tidal current. The total surface motion measured by the Doppler shift of the SAR imagery is comprised of the tidal current, wave orbital motion, stokes drift, wind drift and any other effect that is moving that parcel of fluid. In order to get TSCV, these effects must be accounted for in the WASV and removed from the total surface motion sensed by the Doppler. With regard to Stokes drift - its possible, but would need more work. Its a complicated issue, but briefly the effect will be dwarfed by other contributions to the total surface motion and will be difficult to unpick. This is an interesting thought and we will address this in Section 5.

(Line 369:378) TSCV is defined as the effective mass transport, being composed of any movement of the sea surface that is not accounted for by the WASV, i.e. from the wind (in the case of the GMF of Mouche et al, 2012; or wind and waves in the case of Yurovsky et al, 2019). In this sense, TSCV contains contributions from a wide variety of physical processes that may be isolated depending on the analysis or secondary information used. In the case of the work presented in this paper the TSCV is completely dominated by the contribution from the tide, being orders of magnitude greater than, e.g., Stokes drift or Ekman currents. Being a surface imaging system, OSCAR and the SeaSTAR methodology are also capturing only the ocean motion that is manifest at the surface. The currents sensed by the X-band radar used for comparison are barotropic, as are the currents produced by the MARS2D numerical model (which is depth-averaged). Unravelling baroclinic contributions to the Doppler signal sensed at the surface will require a different experimental design in an area that is not dominated by tidal currents (e.g., the open ocean).

Minor typos: ln's # 45, 67 , 240

(Line 45) 'Sasaki et al., 2014;),' to 'Sasaki et al., 2014),'

(Line 67) 'ATIsystem' removed as replaced with: (Line 64:67) The total surface current vector (TSCV), corresponding to the effective mass transport, contains contributions from surface wind drift, Ekman currents, geostrophic currents and Stokes drift but is dominated by tidal currents in many coastal environments by an order of magnitude. TSCV is derived from the surface motion directly measured by the Doppler shift minus the WASV.

(Line 240) 'jet..' to (Line 261) 'jet.'

Other changes:

Small changes and clarifications aside from those highlighted in this change doc are highlighted as changed in the marked up revised manuscript.

References added:

Anderson, C., Bonekamp, H., Duff, C., Figa-Saldana, J. and Wilson, J.J.W.: Analysis of ASCAT Ocean Backscatter Measurement Noise, IEEE Trans. Geosci. Remote, 50(7), 2449–2457, doi: 10.1109/TGRS.2012.2190739, 2012

Elyouncha, A., Broström, G. and Johnsen, H.: Synergistic utilization of spaceborne SAR observations for monitoring the Baltic Sea flow through the Danish straits. ESS Open Archive, doi: 10.22541/essoar.171466079.98038905/v1, 2024

de Kloe, J., Stoffelen, A. and Verhoef, A: Improved Use of Scatterometer Measurements by Using Stress-Equivalent Reference Winds, IEEE J-STARS, 5, 2340–2347, doi: 10.1109/JSTARS.2017.2685242, 2017

Polverari, F., Wineteer, A., Rodríguez, E., Perkovic-Martin, D., Siqueira, P., Farrar, J. T., Adam, M., Closa Tarrés, M. and Edson, J. B.: A Ka-Band Wind Geophysical Model Function Using Doppler Scatterometer Measurements from the Air-Sea Interaction Tower Experiment, Remote Sens., 14(9), 2067, doi: 10.3390/rs14092067, 2022

Portabella, M. and Stoffelen, A.: Scatterometer Backscatter Uncertainty Due to Wind Variability, IEEE Trans. Geosci. Remote, 44(11), 3356–3362, doi: 10.1109/TGRS.2006.877952, 2006

Wang, Z., Stoffelen, A., Fois, F., Verhoef, A., Zhao, C., Lin, M. and Chen, G.: SST Dependence of Ku- and C-Band Backscatter Measurements, IEEE J-STARS, 10(5), 2135–2146, doi: 10.1109/JSTARS.2016.2600749, 2017.